# Pressurised Intraperitoneal Aerosolised Chemotherapy—Results from the First Hundred Consecutive Procedures

**DOI:** 10.3390/cancers16081559

**Published:** 2024-04-19

**Authors:** David Hoskovec, Zdeněk Krška, Michal Vočka, Soňa Argalácsová, Petr Dytrych

**Affiliations:** 11st Department of Surgery, General University Hospital, 128 08 Prague, Czech Republic; zdenek.krska@vfn.cz (Z.K.); petr.dytrych@vfn.cz (P.D.); 21st Medical Faculty, Charles University, 120 00 Prague, Czech Republic; michal.vocka@vfn.cz (M.V.); sona.argalacsova@vfn.cz (S.A.); 3Department of Oncology, General University Hospital, 128 08 Prague, Czech Republic

**Keywords:** peritoneal carcinomatosis, PIPAC, atomiser

## Abstract

**Simple Summary:**

Peritoneal carcinomatosis is a disease with a very poor prognosis. PIPAC (pressurized intraperitoneal aerosolized chemotherapy) is a promising new technique which can be used to treat patients. We describe our first experience with this oncosurgical technique. We used two different applicators (CAPNOPEN and TOPOL) with similar results. PIPAC is mainly a palliative treatment but it can be used for neoadjuvant therapy. The conversion rate was 12% in our study.

**Abstract:**

PIPAC is a new and promising technique for the intraperitoneal administration of chemotherapy. It can be used in patients with various peritoneal cancer metastases. It is mainly a palliative treatment, but there is some neoadjuvant treatment potential. We have operated on 41 patients with various intra-abdominal cancers. PIPAC was performed every 6 weeks. The indication was extension of peritoneal carcinomatosis beyond the criteria for cytoreductive surgery and HIPEC. The effect was evaluated according to the peritoneal cancer index, the peritoneal regression grading score and the amount of ascites. Complications were classified according to the Clavien-Dindo system. We have performed 100 PIPAC procedures. There were two major complications, classified as Clavien Dindo III (2%). The number of procedures varied from 1 to 6. Five patients switched to cytoreductive surgery and HIPEC, and one was indicated for the watch and wait strategy due to total regression according to PRGS. Three patients are still continuing treatment. The others stopped treatment mainly because of progression of the disease and loss of metastases. We observed a reduction in ascites production soon after PIPAC application. PIPAC is a safe and well-tolerated treatment modality. It is mainly a palliative treatment that can improve the quality of life by reducing the production of ascites, but in about 10% of cases, it can reduce the extent of the disease and allow for further radical treatment.

## 1. Introduction

Peritoneal carcinomatosis is a heterogeneous disease. It has a very poor prognosis and occurs with variable frequency in different oncological diagnoses: 40–50% in ovarian cancer, 15–50% in gastric cancer and 5–15% in colorectal cancer. It is also common in other intra-abdominal and extra-abdominal malignant tumours (tumours of the biliopancreatic system, small intestine, breast, etc.) [1,2]. The occurrence of peritoneal carcinomatosis usually indicates a terminal illness and the standard treatment is palliative chemotherapy or best supportive care. Approximately 50% of patients with PM develop ascites and bowel obstruction in the later stages of the disease [3]. The presence of the plasma–peritoneal barrier, which prevents cytostatic drugs from penetrating the abdominal cavity, limits the efficacy of conventionally administered chemotherapy. To circumvent this barrier, we deliver chemotherapy directly into the abdominal cavity. There are several methods of delivering chemotherapy into the abdominal cavity including hyperthermic intraperitoneal chemotherapy (HIPEC).

Pressurised intraperitoneal aerosolised chemotherapy (PIPAC) is a new and promising technique for the intraperitoneal administration of chemotherapeutic agents. It was developed and described by Prof. Reymond [4] and can be used in cases of advanced peritoneal malignancies, including primary and metastatic cancers of the peritoneal cavity. PIPAC is based on applying a highly concentrated cytostatic aerosol in a gaseous environment (capnoperitoneum) (Figure 1).

The rationale behind PIPAC includes:

(1) Optimising drug distribution using an aerosol rather than a liquid solution;

(2) Applying increased intraperitoneal hydrostatic pressure to increase drug penetration to the targeted tumour tissue;

(3) Limiting blood loss during drug application [5,6].

PIPAC combines the theoretical pharmacokinetic advantages of low-dose intraperitoneal chemotherapy (i.e., low toxicity, high intraperitoneal concentration, and low systemic concentration) with the principles of aerosol (homogeneous intraperitoneal distribution and deeper tissue penetration) [7,8]. This minimally invasive approach with a short postoperative recovery allows PIPAC to be combined with standard chemotherapy.

The procedure is considered safe for both the patient and operating theatre staff, as there is no evidence of contamination of clothing or gloves. Furthermore, no one should be in the operating room during the procedure itself (because of possible accidents, not the risk of contamination) [9,10].

PIPAC is indicated by the intraperitoneal spread of various tumours. The most common are ovarian, gastric, hepatobiliopancreatic, malignant peritoneal mesothelioma and colorectal cancer.

Contraindications are a disorder of GIT patency that include total parenteral nutrition, extensive ascites, and a short expected survival time (this means shorter than several months, because the treatment has to be repeated). Procedures requiring simultaneous GIT sutures/anastomosis are also contraindicated. Relative contraindications include the presence of extraperitoneal metastases, portal vein thrombosis, and poor performance status. Previous cytoreductive surgery with HIPEC may also be a relative contraindication.

Commonly used cytostatic agents are 7.5 mg cisplatin/m^2^ combined with 1.5 mg doxorubicin/m^2^ for most diagnoses, and 92 mg doxorubicin/m^2^ for colorectal cancer [4,11,12,13,14].

PIPAC can be administered in combination with chemotherapy or alone, and is repeated at 4–8 week intervals. A minimally invasive laparoscopic approach, usually with two or three ports, is used to perform the procedure [1,6,15].

## 2. Materials and Methods

All patients who underwent PIPAC at the General University Hospital in Prague were included in the present study. Data on patient characteristics were extracted from the prospectively collected database. Patients were referred to our hospital for intraperitoneal treatment (HIPEC or PIPAC). The type of surgery performed was based on the patient’s medical history, and imaging results (CT, PET-CT, MRI, PET-MRI). The final decision was based on laparoscopic examination, and depending on these results, we continued with HIPEC (usually an open procedure) or PIPAC. The decision was based on the extent of peritoneal disease according to PCI (Sugerbaker’s peritoneal cancer index). PIPAC is performed when the PCI is above 10 in cases of stomach cancer, and above 15 in cases of colorectal cancer and ovarian cancer. For patients unsuitable for extensive cytoreductive surgery and HIPEC, we have lowered this threshold. All patients were treated with systemic chemotherapy prior to PIPAC and between PIPAC procedures.

The procedure was performed under general anaesthesia, and a prophylactic antibiotic injection was administered in accordance with the hospital’s antibiotic policy. The first incision is usually made in the midline (depending on previous scars). We used a Veres needle to insufflate CO_2_. The intraoperative intra-abdominal pressure was 12 mm Hg. After inserting the 10 mm port and camera, we assessed the intra-abdominal findings and under visual control, placed one or two 5 mm ports in the free space. We captured photos, and calculated PCI or sPCI. If there were ascites, we aspirated them, measured the amount, and sent the sample to cytology examination. Biopsies were taken (at least two) and sent to pathology for analysis. A chemotherapeutic aerosol was then applied, and left to take effect for 30 min. The intra-abdominal pressure was maintained at a value of 12 mm Hg. The aerosol was then evacuated via the hospital’s separate waste disposal system. We produced the therapeutic aerosol by CAPNOPEN^®^ (Reger Medizintechnik, GmbH, Villingendorf, Germany). Since 2021, we have used the original Czech device for PIPAC (and PITAC), a mini-invasive surgical atomiser MCR-4 TOPOL^®^, which is patented and has a CE certificate. (Figure 2) For most cancers, we administer 7.5 mg cisplatin/m^2^ body surface and 1.5 mg doxorubicin/m^2^ body surface. For colorectal cancer, we administer 92 mg oxaliplatin/m^2^ body surface. For ovarian and serous peritoneal cancers, we administer carboxyplatin AUC 1.5–3. The procedure is repeated every 6 weeks (+/−1 week).

For the first five cases where the TOPOL micro-atomiser was used, we assessed the safety of the procedure with a new system. We determined the number of particles (Particulate Matter, PM) in the area around the ports (30 cm), 50–70 cm above the patient, and in the area around the waste system. The measurement was repeated three times: before applying PIPAC, immediately after applying PIPAC, and while desufflating the peritoneal cavity.

We used the sPCI (simplified peritoneal cancer index) [16], the amount of ascites and its cytological examination results (positive × negative), and the peritoneal regression grading score (PRGS) [17] to evaluate the effect of the treatment. All procedures are documented by standard photographs (the left and right diaphragm, the subhepatic space, the right and left lower quadrants and the small pelvis). We also used this photographic database to assess the progression or remission of the disease.

The Clavien-Dindo system was used to evaluate postoperative complications [18].

## 3. Results

Since 2020, we have operated on 41 patients, including 19 male and 22 female patients. The mean age of the patients was 58 years (13–78 years). The most common diagnosis was colorectal cancer with 17 cases. Gastric cancer was the second most common with 10 patients (Table 1). The number of PIPAC procedures ranged from 1 to 6 (Figure 3). In all but two cases, the postoperative course was uneventful. The majority of patients were discharged on the second or third post-operative day.

We determined that the procedure was safe for patients and staff even when the TOPOL atomiser was used. We found no difference in the number of foreign particles before, during or after the procedure, not near the ports, above the patient, or near the evacuation system. (PM 2.5 was 2.5–3.1 µg/m^3^ and PM 10 was 2.7–3.3 µg/m^3^ in all locations, regardless of PIPAC application).

Two serious complications (2%) occurred in our patients (Clavien-Dindo 3a and 3b). In the first case, a patient with gastric cancer showed signs of irritation of the peritoneum and pneumoperitoneum on a CT scan three days after the third PIPAC. He was admitted to emergency surgery and we found a perforation of the large bowel. Due to the extensive carcinomatosis located around the ileocecal region (as well as in the entire abdomen), resection was considered difficult. Therefore, we sutured the colon and created a diversion ileostomy. The bowel perforation was located at the metastasis site. The second patient who developed a complication was indicated for PIPAC due to peritoneal metastases of colorectal cancer. Obesity and limited intraperitoneal space made the procedure difficult. There was a rise in temperature and CRP in the postoperative course, but the CT scan did not show any intra-abdominal complications. The condition improved after antibiotic treatment. Two weeks later, there was a recurrence of fever and high CRP (the patient was discharged earlier). An intra-abdominal abscess was found and successfully treated with antibiotics and CT-navigated drainage.

Out of 14 patients, we performed only one PIPAC procedure. Two of the patients had indications for cytoreductive surgery and subsequent HIPEC. In these cases, the extent of the intraperitoneal spread of the carcinomatosis reached the threshold for the PIPAC procedure. In one case, which will be discussed in detail, we did not find any peritoneal seeding during the procedure. This was a thirteen-year-old patient with colorectal cancer. The most common reason for discontinuing PIPAC was the loss of distribution space.

### 3.1. Colorectal Cancer

There were 17 patients with colorectal cancer (12 males and 5 females). Most of them had previously undergone surgery with radical or palliative intent, and all had previously received chemotherapy. The number of PIPAC treatments varied from 1 to 6. The chemotherapeutic regimen was 92 mg oxaliplatin/m^2^ body surface area for 30 min. The reason for discontinuing PIPAC treatment was the loss of distribution area in four cases. Only one application was possible in three patients. The second attempt was unsuccessful because we could not find a free intra-abdominal space and establish a capnoperitoneum. Otherwise, there was only a small space that was not suitable for aerosol treatment. One such case involved a patient with an abdominal abscess, as mentioned above. Two patients completed three and four PIPAC procedures, respectively. We ceased treatment due to a loss of distribution space. Two patients underwent surgery due to bowel obstruction after the first and third PIPAC procedures. We did not continue with intra-abdominal chemotherapy. One patient refused to continue after receiving the second PIPAC. We found extra-abdominal disease progression in three patients after two PIPAC cycles for one patient and after five cycles for two others. In all three cases, the intra-abdominal extension of the disease was stable. There was no progression of the disease, stable PCI, or sPCI. The PRGS was 3 or 2 at the time of the last treatment. Newly diagnosed lung metastases were the reason for discontinuing PIPAC in all of these cases. Six patients had ascites prior to treatment. Ascites production decreased significantly after the first PIPAC, and this effect persisted throughout treatment. We conducted four cycles in one patient (70 years old female) with large peritoneal involvement (PCI 23, sPCI 13). The macroscopic extent of carcinomatosis remained the same in all cycles, but the peritoneal regression grading score was 1 after the second treatment. Despite multiple biopsies during the last two PIPACs, the pathologist found no cancer cells and confirmed a PRGS score of 1. The MDT decision was to stop intraperitoneal chemotherapy and only conduct follow up.

The youngest patient in our group was a 13-year-old male. He had a right hemicolectmyfor intestinal obstruction due to cancer in a hepatic flexure. During surgery, peritoneal spread occurred in the right lower quadrant and after adjuvant chemotherapy, PIPAC was indicated. We performed a laparoscopy and found no peritoneal seeding. We applied PIPAC as an adjuvant treatment, and the paediatric oncologist continued with chemotherapy.

In the last patient, surgery was indicated for liver metastases after sigmoid resection. However, peritoneal carcinomatosis was found, and only an exploratory laparotomy was performed. We performed laparoscopy and PIPAC, but the sPCI was low—3. Therefore, we performed a peritonectomy, omntectmy, left liver lobe resection and HPEC 8 weeks after the laparoscopy and PIPAC.

### 3.2. Gastric Cancer

Patients with stomach cancer were the second most common group in our study. There were 10 patients: 5 males and 5 females. We used 7.5 mg cisplatin/m^2^ body surface and 1.5 mg doxorubicin/m^2^ body surface in all cases. Radical surgery was attempted prior to PIPAC in only three cases—total gastrectomy plus D2 lymfadenctomy, in one case combined with HIPEC. The interval between surgery and PIPAC varied from 8 to 19 months. Two patients were treated with palliative HIPEC prior to PIPAC. One patient was the first patient in this study and we used HIPEC twice before PIPAC was available in our department. The second patient underwent laparoscopic HIPEC outside our hospital and was later indicated for repeat PIPAC. His treatment is still ongoing. The patients underwent a maximum of three courses of PIPAC before we discontinued treatment, usually due to disease progression or the loss of distribution space. The extent of intra-abdominal disease remained stable in most patients, and there was no disease regression according to PRGS. However, we saw a reduction in ascites production after the first PIPAC cycle in six patients. Disease progression was usually outside the abdominal cavity.

### 3.3. Primary Peritoneal Cancer

We operated on six female patients with primary high-grade peritoneal cancer. The mean age was 66 years (42–77 years). All had primary surgery in different gynaecological departments, and all were referred to our department due to intra-abdominal tumour recurrence. We used carboxyplatin AUC 1.5–3 in these cases. The number of PIPAC procedures varied from three to six. One patient refused further treatment using this technique after the third PIPAC, despite a reduction in sPCI after the first PIPAC. We stopped PIPAC treatment in three cases due to disease progression in the abdominal cavity and liver metastases. One patient was indicated for surgery and HIPEC after the sixth procedure. There was a decrease in sPCI: the initial value was 10 and decreased to 6 during the last PIPAC according to sPCI. Surprisingly, the PRGS score was unchanged at 4, which indicated no effect. However the decrease in carcinomatosis extent was clear (Figure 4 and Figure 5). Unfortunately, there was also tumour progression in the omental bursa according to preoperative PET-CT. We found that the tumour surrounded the celiac trunk and its branches (except for involvement of both diaphragms); therefore, radical surgery was impossible. Interestingly, there was sPCI regression in this group of patients, but the PRGS score was usually unchanged. The last patient, a 77-year-old woman with diaphragm, small pelvis and small bowel involvement according to preoperative PET-CT, was indicated for PIPAC due to her age and an expectedly high sPCI. Surprisingly, we found only minimal disease (sPCI 1), and she was indicated for CRS + HIPEC.

### 3.4. Mesothelioma

We treated five patients with mesothelioma: three females and two males. The mean age of these patients was 60 years (41–72 years). We used 7.5 mg/m^2^ body surface area of cisplatin and 1.5 mg/m^2^ body surface area of doxorubicin in all cases. All patients except one were diagnosed during surgery, whether a diagnostic laparoscopy or emergency surgery (laparoscopic appendectomy, bowel obstruction). The last patient was BRCA 1-positive. She suffered from pleural mesothelioma and was diagnosed with intra-abdominal disease during follow-up four years after the initial surgery. We performed a diagnostic laparoscopy and PIPAC, but the sPCI was low, at around 6. Therefore, we proceeded with cytoreductive surgery, peritonectomy and HIPEC 8 weeks later. The other five patients had 3–5 PIPAC procedures. We discontinued PIPAC and continued with chemo-biotherapy because the disease was stable, without sPCI progression, but without regression according to PRGS (3–4). In one case, we indicated cytoreductive surgery because the patient had significant omental involvement outside the parietal and visceral peritoneum and high ascites production (about 6–8 litres between PIPAC procedures). Therefore, we decided to perform laparoscopic omentectomy and palliative HIPEC to reduce malignant ascites production and continue with oncological chemo-biotherapy. In short, in cases of mesothelioma, we could prevent disease progression in three cases. In one case, we switched to radial CRS + HIPEC, and in the last case, we switched to palliative CRS + HIPEC.

### 3.5. Other Indication

We operated on two patients with gallbladder carcinoma. In both cases, a laparoscopic cholecystectomy was performed, and gallbladder carcinoma was found incidentally. In both cases, the second operation involved the excision of the gallbladder bed and lymphadenectomy (hepatoduodenal ligament, around hepatic artery and retroduodenal). Intraperitoneal metastases were found one and two years after surgery, and both patients were indicated for PIPAC. We used cisplatin + doxorubicin in both cases. We stopped this treatment after the first and third application due to disease progression.

The final diagnosis was intra-abdominal metastases of the neuroendocrine tumour. The patient suffered from liver cirrhosis and liver failure and was indicated for liver transplantation. During surgery, small peritoneal implants of the neuroendocrine tumour were found. Transplantation was contraindicated, as was oncological therapy due to liver function. The multidisciplinary team decided on PIPAC with cisplatin. The extent of metastases was relatively low, around sPCI 4, and there were between 1500 and 2000 mL of ascites (cytology was always negative). The treatment was without complications, but without success (PRGS 4, stable ascites amount, sPCI 4). We stopped the treatment before the third application.

## 4. Discussion

PIPAC is a new treatment modality still under evaluation. It is an emerging technique for the treatment and palliation of peritoneal metastases in various cancers. Some studies show that it is one of the best methods to manage the burden of advanced intraperitoneal metastases by reducing or halting disease progression and improving the quality of life [19]. We summarise our experience and results with the first one hundred consecutive PIPAC procedures. This study summarises the first experience of pressurised intraperitoneal chemotherapy in the Czech Republic. To our knowledge, this is the first PIPAC study using a different applicator than CAPNOPEN^®^ (Reger Medizintechnik, GmbH, Villingendorf, Germany). For most of these procedures, we used an original Czech device—the MCR-4 TOPOL^®^atomiser—developed and manufactured by Skala Medica Inc.

Patients indicated for PIPAC treatment suffer from peritoneal dissemination of various cancer types. Peritoneal seeding of the tumours was always the only site of metastasis, and patients with extraperitoneal metastases were contraindicated for this treatment. The main disease entities were ovarian, gastric, colorectal, peritoneal mesothelioma and other cancers, including those of hepatobiliary and pancreatic origin. [13] This is similar to our study. We do not indicate patients with pseudomyxoma peritonei for PIPAC treatment; rather, patients are indicated for cytoreductive surgery and HIPEC and are treated with this approach. The sporadic indication not yet described in the literature is PIPAC application for the treatment of neuroendocrine tumours. Liver function contraindicated further oncological treatment and intraperitoneal spread of NET contraindicated transplantation in our patient.

Disease extent was the main criterion for HIPEC or PIPAC indication depending on the diagnosis. We performed diagnostic laparoscopy in equivocal cases for a definitive decision. Performance status was the second criterion to stratify patients into PIPAC or CRS + HIPEC treatment.

The PIPAC technique and chemotherapeutic regimens are generally standardised [12,20,21], and we follow these guidelines. Only in serous peritoneal carcinoma have we used carboxyplatin. Our multidisciplinary team and our pharmacologist discussed this decision extensively and accepted its outcome.

One of the most important things is to correctly assess the disease extent and changes during treatment. We use three main criteria: the peritoneal cancer index, the peritoneal regression grading score (PRGS), and the amount of ascites, including cytology. We switched to the simplified peritoneal cancer index instead of Sugerbaker’s PCI because it was easier to calculate during laparoscopy. Furthermore we believe that sPCI is an adequate scoring system to assess macroscopic changes in peritoneal spread. Proper documentation of the disease is an important issue too. We use standardised photography of the left and right diaphragm, subhepatic space, right and left lower quadrant, and small pelvis. The operating surgeon can access these photos from the hospital’s online database, which is also available in the operating theatre. For any given patient, we try not to change the operating surgeon. PRGS is the most widely used system for assessing microscopic disease regression [12]. This is simple for pathologists and coherent for surgeons and oncologists.

Intraperitoneal aerosol chemotherapy results can be evaluated from different perspectives. We must always bear in mind that these patients suffer from advanced oncological disease and, in many cases, end-stage disease. PIPAC is an invasive treatment administered in the operating theatre and requires general anaesthesia. The appropriate endpoints to evaluate or promote this technique could be one or more of the following: hospital stay length, rate of major complications, quality of life (and patients’ willingness to undergo repeat surgery), and morbidity [13,22].

Most of our patients were discharged on the second, maximum third postoperative day. Postoperative recovery is rapid, and all but one of our patients agreed to continue treatment. This finding is indirect proof of good tolerance and minimal changes to patients’ quality of life. We had only two serious complications, Clavien Dindo III, which equates to 2%. This is similar to other authors’ published PIPAC treatment results [6,12,19,20].

Reduced ascites production can improve our patients’ quality of life. We have 14 patients with significant ascites (above 1000 mL) at the institution of PIPAC treatment. Thirteen patients experienced reduced ascites production, usually after the first PIPAC. This is probably common in both intraperitoneal chemotherapies (HIPEC and PIPAC) [23,24,25]. On the other hand, two patients were without ascites, or with a small amount of it, but they developed ascites during treatment.

The oncological effects of the treatment are described by different outcomes: PRGS, sPCI, extra-abdominal disease progression, and a loss of intraperitoneal space, which prevent further PIPAC treatment. Disease regression, which allows for the continuation of CRS and HIPEC, is the most remarkable effect that deserves special attention. It can be expected in 7–17.8% of cases [7,19,26,27]. In this study, we see this result in five of our patients (12%). Interestingly PRGS and sPCI changes were not parallel. We observed patients with clear sPCI regression and stable, minimal changes in PRGS (most of these patients belong to the group with primary peritoneal carcinoma). Conversely, we had patients with macroscopically stable disease according to sPCI, but with significant responses according to PRGS, including a complete response in the peritoneal regression grading system (PRGS 1). This difference between macroscopic and microscopic response has not been described in the literature. Exploratory laparotomy and CRS/HIPEC were offered to three patients after their first PIPAC treatment because they had limited and resectable peritoneal metastases at exploration. These patients actually had no indication for PIPAC, but from the preoperative examination (or description of the previous surgery outside our department it was not clear), they could probably have been recommended for upfront CRS and HIPEC.

The number of PIPAC procedures required to re-evaluate the efficacy of this treatment is unclear. Although we conducted the first assessment after three cycles, five cycles is probably more appropriate. PIPAC should be stopped if there is no response. In the literature, the number of cycles varies from 3 to 6 [19,28,29].

We stopped PIPAC treatment in seven patients due to a loss of distribution space, in thirteen patients due to disease progression, in six patients due to non-response, and in two due to regression (switch to CRS + HIPEC or watch and wait). These findings are consistent with data published in the literature [30,31].

## 5. Conclusions

In conclusion, PIPAC can be considered a safe and promising alternative treatment for patients with advanced isolated refractory peritoneal disease. We did not find any differences in the peri- and postoperative courses according to the devices used (CAPNOPEN and TOPOL). PIPAC is a palliative treatment in the majority of patients and is well tolerated with a reasonable quality of life. Many patients showed reduced malignant ascites after the first PIPAC treatment. It is necessary to reach a consensus on the appropriate time for PIPAC treatment efficiency re-evaluation. In our opinion, re-evaluation should occur during the fifth PIPAC cycle. We must count all variables (sPCI, PRGS, ascites, cytology) because they do not change together. Excluding extra-abdominal spread of the tumour is a sine qua non. Special emphasis should be placed on groups with a complete pathological response or patients indicted for subsequent CRS + HIPEC.

## Figures and Tables

**Figure 1 cancers-16-01559-f001:**
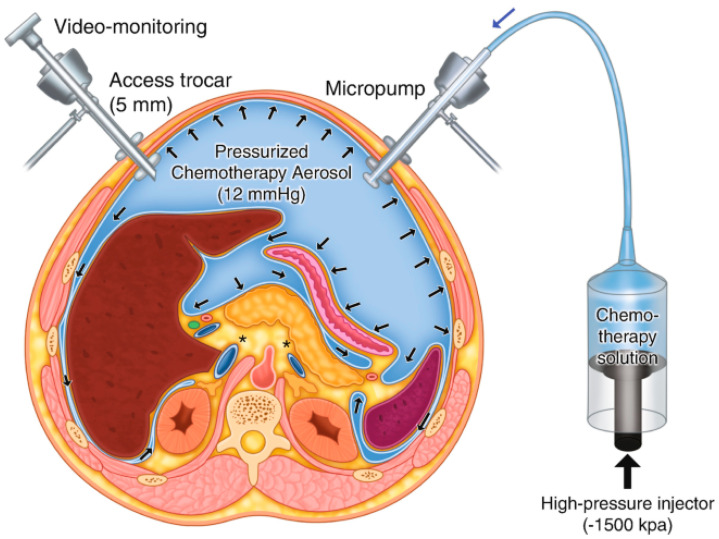
Scheme of PIPAC [4].

**Figure 2 cancers-16-01559-f002:**
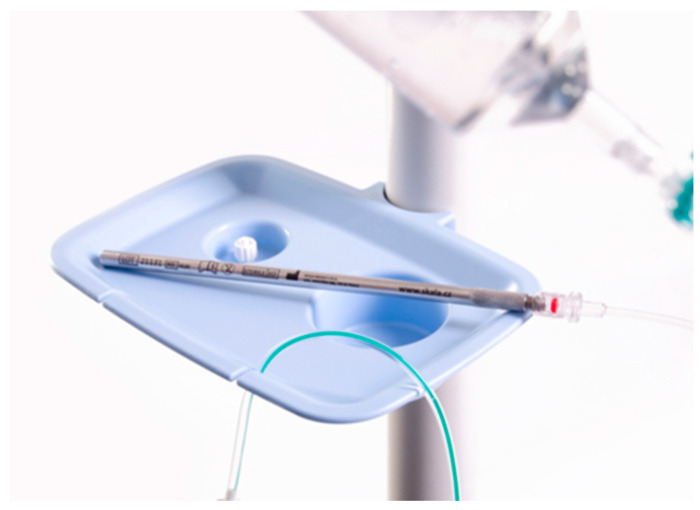
Mini-invasive surgical atomiser MCR-4 TOPOL^®^.

**Figure 3 cancers-16-01559-f003:**
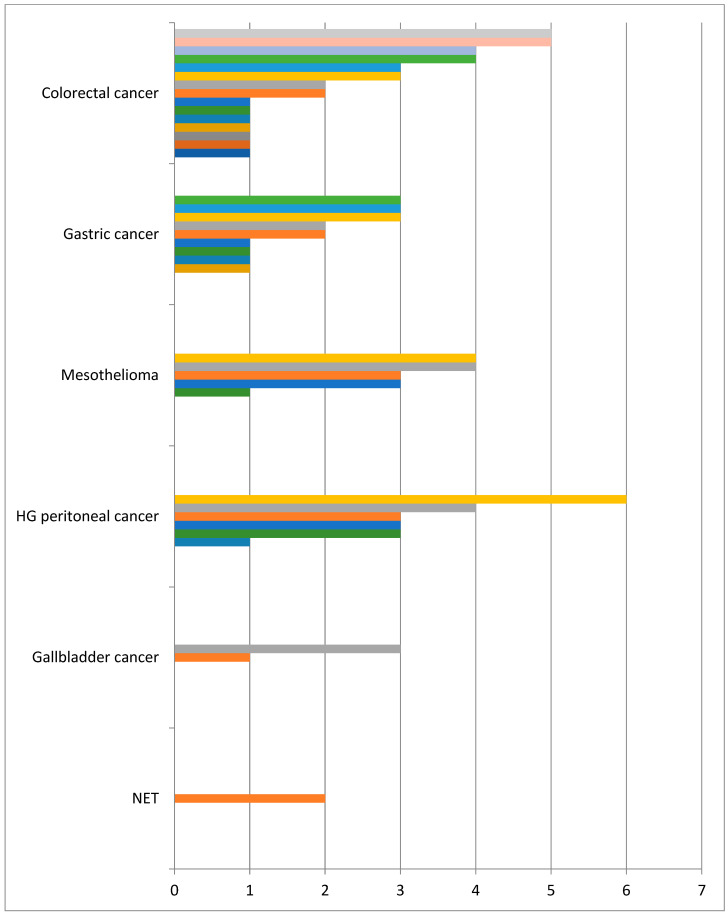
No of PIPAC in finished treatment.

**Figure 4 cancers-16-01559-f004:**
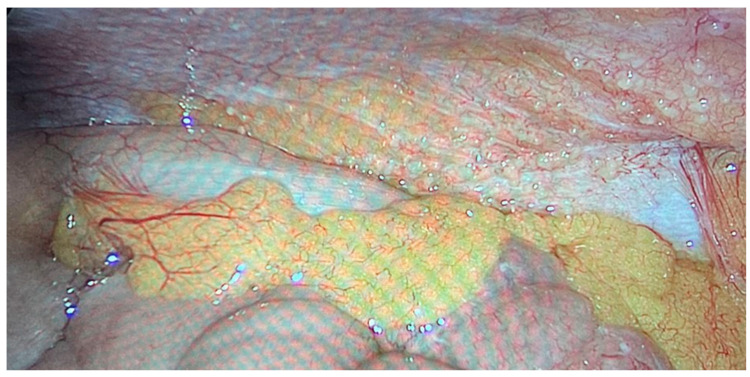
Primary peritoneal cancer PIPAC 1.

**Figure 5 cancers-16-01559-f005:**
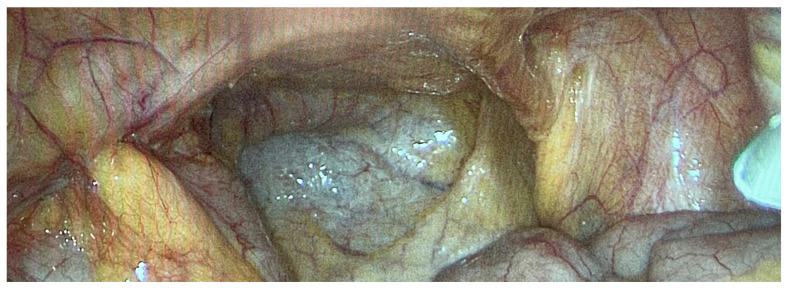
Primary peritoneal cancer PIPAC 6.

**Table 1 cancers-16-01559-t001:** Patients characteristics.

Diagnosis	Male	Female	Age
Colorectal	12	5	55.2 (13–78)
Gastric	5	5	57.3 (37–74)
HG peritoneal		6	66.6 (42–77)
Mesotelioma	2	3	60 (41–74)
Gallbladder		2	59 (51–67)
NET		1	59
**Total**	**19**	**22**	**58.2 (13–78)**

## Data Availability

Data are contained within the article.

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
