# Peer review of "Pressurised Intraperitoneal Aerosolised Chemotherapy—Results from the First Hundred Consecutive Procedures"

_cancers, 2024, doi:10.3390/cancers16081559_

Round 1

Reviewer 1 Report

Comments and Suggestions for Authors

-Raw 73 pag 3: ..as palliative treatment, a short life expectancy is not uncommon, better remove the sentence or clarify..

-Personal note: the oolicy of patients selection is very strict: this is a good thing to me, meaning you don't over-extend the indications for the procedures you perform.

-Raw 99 pag 4: personal note: why the first incision in the middle? If previous scars with adhesions/PC are present, it is easier to have misunderknown bowel lesions/bleeding.. I perform single port access, with 4 cm incisioni i see what happens..

-Figure 3 pag 6: mestelioma insteado of mesothelioma in the figure

-Raw 140-142 page 6: General note: you confront 2 different devices and write no differences in foreign particles were found into the 2 systems. If you provide a tab or a graphic of those datas, this statement you made will be supported better than "because I say so"...  

Author Response

Dear Reviewer,

Thank you very much for your review and suggestions. We have tried to do our best to address your comments. They are in the text. We have clarified the term "short expected survival time" as shorter then several months for more cycles of PIPAC than one or two. About the first incision, it depends on the previous procedures, but if it is possible, we start in the midline. Your suggestion with single port is very interesting and thank you for it. But in our hospital single port is not widely used, so this is probably the reason why we do not use it. We added some data on measuring potential contamination, but we have not compared two devices. We only did a checks when we switched to new PIPAC device - for our own safety and that of our patients.

And thank you for your sentence about our indication criteria.

Best regards

Reviewer 2 Report

Comments and Suggestions for Authors

Regarding the article: PIPAC – Results from the first hundred consecutive procedures

The study is promising for patients with carcinomatosis.

The authors state:

The procedure is considered safe for both the patient and operating theatre staff, as  there is no evidence of contamination of clothing or gloves. Furthermore, no one should  be in the operating room during the procedure itself (because of possible accidents, not  the risk of contamination). [7]

If it s safe why leave the room in fear of accidents?

Several sentences have to be rephrased. Among them:

 In our study was the conversion rate 12 %. Row 12

PIPAC can only be applied once to three patients. Row 169

The youngest patient in our group, a 13-year-old male, underwent surgery for intestinal obstruction due to cancer in a hepatic flexure and right hemicolectomy. Row  189

Arithmetic problems: There were 10 patients: 5 males and 7 females… Row 202

Reference 18 faulty positioning and content:

18. . Hoskovec, D.; Varga, J.; Dytrych, P.; Konecna, E.; Matek, J. Peritoneal Lavage Examination as a Prognostic Tool in Cases of Gastric Cancer. aoms 2017, 3, 612–616, doi:10.5114/aoms.2016.64044.

Hoskovec, D., Varga, J., Dytrych, P., Konecna, E., & Matek, J. (2017). Peritoneal lavage examination as a prognostic tool in cases of gastric cancer. Archives of medical science : AMS13(3), 612–616. https://doi.org/10.5114/aoms.2016.64044

Otherwise a local premiere with a new device.

Author Response

Dear Reviewer,

Thank you very much for your review and suggestions. We have tried to do our best to address your comments. They are in the text. The recommendation to leave the room is repeated in the literature, although we consider the procedure to be safe. That's why we mentioned it. We have rephrased some sentences according to your suggestions and, of course, corrected the number of patients where there was a mathematical error.

Best regards